# LA-BALD: An Information-Theoretic Image Labeling Task Sampler

## Abstract

Large-scale visual recognition datasets with high-quality labels enable many computer vision applications, but also come with enormous annotation costs, especially since multiple annotators are typically queried per image to obtain a more reliable label. Recent work in label aggregation consolidates human annotations by combining them with the predictions of an online-learned predictive model. In this work, we devise an image labeling *task sampler* that actively selects image-worker pairs to efficiently reduce the noise in the human annotations and improve the predictive model *at the same time*. We propose an information-theoretic task sampler, Label Aggregation BALD (LA-BALD), to maximize the information contributing to the labeled dataset via human annotations and the model. The simulated experiments on ImageNet100-sandbox show that LA-BALD reduces the number of annotations by 19% and 12% on average compared to the two types of baselines. Our analysis shows that LA-BALD provides both more accurate annotations and better online-learned predictive model, leading to a better labeling efficiency over the baselines.

## 1 Introduction

Machine learning has led to large advances in a wide range of applications such as machine translation, early cancer detection, virtual reality, and autonomous driving. Large-scale labeled datasets play a vital role in the success of modern ML. The CheXNet dataset Rajpurkar et al. (2017) benefits automatic chest radiograph interpretation by providing clinical decision support. The Waymo Open dataset Sun et al. (2020) advances machine perception for self-driving vehicles by collecting data from diverse geographic locations. Diverse animal datasets Beery et al. (2018); Swanson et al. (2015) facilitate automatic animal monitoring.

Human annotators play an essential role in large-scale and high-quality dataset creation Vaughan (2018). The level of inter-human variabilities, such as workers' interest and familiarity with the topic, and perceived task difficulty Kazai et al. (2012), are major factors that govern dataset quality Giuffrida et al. (2018); Jungo et al. (2018). However, the inter-human variability is a double-edged sword for data labeling. With an unlimited monetary budget, we can approximate true label distributions by sampling multiple diverse workers per each example Peterson et al. (2019). On the other hand, we might obtain high-variance noisy labels when we operate with a limited budget and thus need to trust a single or very few annotators per example.

Given the collected human annotations, label aggregation is a common way to infer the latent true labels Zheng et al. (2017). Usually, the aggregator captures each annotation's quality by estimating the data difficulty and the workers' competencies. Recent success in label aggregation for image labeling Branson et al. (2017); Liao et al. (2021) leverages data similarity via an online-learned predictive model to infer the true labels, increasing the labeling efficiency by a large margin. Under this framework, an aggregator's label quality is blocked by the individual annotation quality and the predictive model's performance.

In this work, adopting the state-of-the-art label aggregator Liao et al. (2021), we propose an information-theoretic image labeling task sampler, **L**abel **A**ggregation **BALD** (LA-BALD), targeting on both blockers: *i)* which image-worker pair provides the best expected quality, and *ii)* which image labels benefit the predicted model the most. We formulate two different goals into an information maximization problem. As shown in Fig. 1, each annotation provides information to the

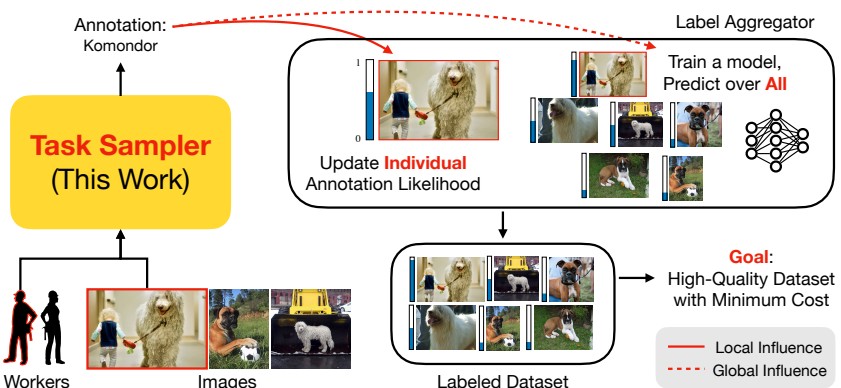

Figure 1: **Goal of an Image Labeling Task Sampler.** State-of-the-art label aggregator Liao et al. (2021) leverages an online-learned predictive model to increase the labeling efficiency. In this framework, each annotation influences the labeled dataset both *locally* (via individual human annotation) and *globally* (via the online-learned predictive model). The image labeling task sampler needs to maximize the information contributing to the labeled dataset via the local and global influence by choosing image-worker pairs in an iterative fashion.

labeled dataset in two ways: local and global influences. The local influence is associated with the individual label, and the global influence relates to the online-learned predictive model. Our proposed task sampler maximizes the expected information contributing to the labeled dataset via both the local and global influence by choosing relevant image-worker pairs.

We provide rigorous empirical comparisons between LA-BALD and the baselines on ImageNet100-sandbox Liao et al. (2021); Tian et al. (2020) and show that LA-BALD reduces the number of annotations by 19% and 12% on average compared to two different types of baselines. Our analysis shows that LA-BALD provides both more accurate annotations and a better online-learned predictive model, resulting in the improvements over the baselines. [1]

## 2 RELATED WORK

### 2.1 LABEL AGGREGATION

Crowdsourcing provides low-cost noisy annotations. To infer true labels from noisy annotations, one must infer each annotation's importance. The seminal work on this problem, dating back 40 years, is the Dawid-Skene model Dawid & Skene (1979) (DS model), optimized with EM . Many variants Demartini et al. (2012); Kim & Ghahramani (2012); Ma et al. (2015); Van Horn et al. (2018) adapt the DS model to different settings. GLAD Whitehill et al. (2009) additionally models heterogeneous image difficulty. Welinder et al. (2010) uses multidimensional entities to represent workers with different competencies and biases. EBCC Li et al. (2019) scales to larger worker cohorts by modelling worker correlations. However, as studied in the survey by Zheng et al. (2017), there is no dominant label aggregation algorithm across all datasets. Recent trend Branson et al. (2017) in label aggregation combines manual labelling with automatic labelling, opening up a new research paradigm. In this work, we adopt the label aggregator proposed by Liao et al. (2021) because of its thorough study of the different factors and improved labelling efficiency.

### 2.2 ACTIVE LEARNING

Active learning aims to increase the ML model performance and to reduce the labelling cost by actively sampling labels from the oracle Settles (2009). Recent work on active learning for neural networks approaches the problem by core-set construction Sener & Savarese (2018), information-theoretic scores Houlsby et al. (2011) with dropout mask Gal et al. (2017); Kirsch et al. (2019), or

---

[1]The code is released in the anonymized repository at https://anonymous.4open.science/r/LA-BALD-8B7D/README.md

reinforcement learning Fang et al. (2017). Unlike active learning, in image labelling, we need to consider heterogeneous workers and data re-labelling.

**Heterogeneous workers** yield noisy annotations. Prior work mitigates this problem by training a different classifier Zhang & Chaudhuri (2015) to predict where weak workers differ from strong workers, leveraging abstention responses Yan et al. (2016) to avoid noisy observations, or filtering them out Younesian et al. (2020). CEAL Huang et al. (2017) and ABL Gao & Saar-Tsechansky (2020) further incorporate worker costs into consideration. **Re-Labeling** has been a common practice in large-scale dataset annotation Lin et al. (2014). Prior work Ipeirotis et al. (2012) recommends re-labelling on a selected subset when annotations are imperfect. Re-active learning Lin et al. (2016) computes the expected error reduction to the contribution of possibly repeated annotations.

Most related work to ours are Yan et al. (2011; 2012b) which perform image-worker sampling to increase the model accuracy via an information-theoretic acquisition function. Unlike the learning algorithm used in prior work Yan et al. (2012b), our label aggregator learns from the worker annotations and combines them with worker annotations in a probabilistic manner. In addition, in image labelling, the target goal is to increase the label accuracy, not the accuracy of a novel test set. To summarize, prior work Yan et al. (2012b) only considers the global influence described in Fig. 1.

## 3 BACKGROUND

In this work, we propose an information-theoretic task sampler for image labelling. In this section, we formulate the problem and introduce the notation and the essential background knowledge of the label aggregator in the prior work Liao et al. (2021). For more details on the label aggregator, please refer to Liao et al. (2021).

### 3.1 PROBLEM FORMULATION

We target the image labelling of a $K$-class image classification problem. Given $N$ images $X = \{x_i\}_{i=1:N}$ and $M$ workers $\{j\}_{j=1:M}$ with unknown skills $W = \{w_j\}_{j=1:M}$, we infer the latent ground truth $Y = \{y_i\}_{i=1:N}, y_i \in [K]$. We represent each worker's skill $w_j$ as a $K$-class confusion matrix.

We formulate the task sampling problem as an active learning problem over images and workers. An acquisition function $S(.)$ is defined over images and workers. At each step $t$, the task sampler samples a batch $B$ of image worker pairs according to the acquisition function $S$. For each image-worker pair $(i, j)$, we obtain an annotation $z_{ij}$. In Sec. 3.2, we explain how to infer the latent ground truth $Y$ from the annotations $Z$.

### 3.2 LABEL AGGREGATION

Label aggregation consolidates noisy annotations. The state-of-the-art label aggregator Liao et al. (2021); Branson et al. (2017) follows the classic Dawid-Skene (DS) model Dawid & Skene (1979) and performs expectation-maximization (EM) to infer latent ground truths. In the DS model, the generation of noisy annotations $Z$ is conditioned on both the latent ground truth $Y$ and the latent worker skills $W$, as visualized in Fig. 2. We can express the joint probability as $p(Z, Y, W) = p(Z|Y, W)p(Y)p(W)$. Let the annotations before step $t$ as $Z_t$ and $\hat{Y} = \{\hat{y}_i\}_{i=1:N}, \hat{W} = \{\hat{w}_j\}_{j=1:M}$ be the maximum likelihood solution. At step $t$, we can write down the E-step and M-step via the Bayes rule in Eq. 1.

We adopt the label aggregator in Liao et al. (2021) that incorporates the predictions of an online-learned predictive model with DS model Branson et al. (2017); Liao et al. (2021) results in drastic improvements. The predictions $p_\theta(Y|X)$ are treated as the prior for latent ground truths $Y$ as in Eq. 2. The online-learned predictive model is trained on the estimated labels at the previous step $p(Y|Z_{t-1}, X)$. We defer the model training details to the appendix.

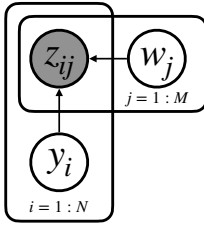

Figure 2: **Graphical Model in the DS Model Dawid & Skene (1979).** We use $i$ to index images and $j$ to index workers. In the DS model, the annotation $z_{ij}$ is conditioned on the ground truth $y_i$ and the worker skills $w_j$.

**Algorithm 1** Image Labeling Process

1: **function** TASKSAMPLER
2:     **Input:** Unconfident set $U$
3:     Compute $S(i,j), \forall (i,j), i \in U, j \in [M]$
4:     $\{(i_1,j_1),...,(i_B,j_B)\} \leftarrow$ Top $B$ $(i,j)$ pairs in $S$
5:     **return** $\{(i_1,j_1),...,(i_B,j_B)\}$
6: **end function**
7: **Input:** Images $X$, threshold $\gamma$
8: $Z_0 = \emptyset, t = 1$
9: $\hat{Y}, \hat{W} \leftarrow$ Label Aggregator$(X, Z_0)$ // Eq. 2
10: **repeat**
11:     $U \leftarrow \{i | \arg\max_{k \in K} \hat{y}_i \leq \gamma, \forall \hat{y}_i \in \hat{Y}\}$
12:     $\{(i_1,j_1),...,(i_B,j_B)\} \leftarrow$ TaskSampler$(U)$
13:     $Z_t \leftarrow \{Z_{t-1}, \text{Workers}(\{(i_1,j_1),...,(i_B,j_B)\})\}$
14:     $\hat{Y}, \hat{W} \leftarrow$ Label Aggregator$(X, Z_t)$ // Eq. 2
15:     $t \leftarrow t + 1$
16: **until** $|U| = 0$

E-step: $p(Y|Z_t) \propto p(Y)p(Z_t|Y,\hat{W})$

M-step: $p(W|Z_t) \propto p(W)p(Z_t|\hat{Y},W)$    (1)

E-step: $p(Y|Z_t, X) \underset{\sim}{\propto} p_\theta(Y|X)p(Z_t|Y,\hat{W})$

M-step: $p(W|Z_t) \propto p(W)p(Z_t|\hat{Y},W)$

$$\theta \leftarrow \arg\min_{\theta'} \sum_i \mathcal{L}(x_i, p(y_i|Z_{t-1}, X), \theta')$$

(2)

For a $K$-class image classification problem, at each step $t$, we first determine the image set that requires annotations, the unconfident set $U$, constructed by $\gamma$-thresholding the max probability of the estimated labels $\hat{y}_i$ over $K$ classes. We sample $B$ image-worker pairs according to the acquisition function $S$ overall workers and the unconfident set $U$. Finally, we perform label aggregation in Eq 2 with the annotations $Z_t$. Algo 1 shows the pseudocode of the online image labelling process.

### 3.3 BAYESIAN ACTIVE LEARNING BY DISAGREEMENT (BALD)

In this work, we formulate the problem as an active learning problem that maximizes the information gained from the labelled dataset. Here, we give a brief introduction to applying information-theoretic acquisition functions in active learning. BALD Houlsby et al. (2011) is an information-theoretic sampler for active learning, measuring the mutual information between ground truth $Y$ and model parameters $\theta$:

$$S_{\text{BALD}} = \mathbb{I}(y_i; \theta) = \mathbb{H}(y_i) - \mathbb{H}(y_i|\theta)$$

(3)

where $\mathbb{I}, \mathbb{H}$ denote mutual information and entropy. Applying BALD to deep neural networks is challenging since it requires computing the distribution of millions of parameters. Prior works Gal & Ghahramani (2016); Gal et al. (2017) use MC dropout to perform variational approximation. Recent works Kirsch et al. (2021); Mindermann et al. (2021) extend BALD to training examples prioritization and distribution shifts between unlabeled sets and test sets, respectively. We defer the details of MC dropout to the appendix. In this work, we extend BALD to image labelling.

## 4 METHOD

Our goal is to improve labelling efficiency by actively sampling image-worker pairs. Since the label aggregator generates labels, we need to consider how the label aggregator processes the annotations when designing our image labelling task sampler. In particular, we target the label aggregator discussed in Sec. 3.2. The main challenge is quantifying the improvements of the online-learned model and the improvements through EM under the same framework.

We devise a novel information-theoretic acquisition function, LA-BALD, to sample image-worker pairs. LA-BALD is defined over image-worker pairs and explicitly considers the information gains to the labelled dataset in two ways: local and global influences. In this section, we start by inspecting how the labels change after getting a single annotation in Sec. 4.1 and propose LA-BALD in Sec. 4.2. Finally, we analyze the bias of LA-BALD. We provide additional analysis in the appendix.

### 4.1 LABELS CHANGE FROM A SINGE ANNOTATION

We adopt the label aggregator described in Sec. 3.2. By adding an annotation $z_{ij}$, the annotation likelihood is expanded to $p(z_{ij}, Z_t | Y, W)$. Tracing the exact change of adding an arbitrary annotation in $Y, W$ is computationally expensive. Therefore, we make the following assumption:

**Assumption 1.** *By adding an annotation $z_{ij}$, the change of the estimated worker skills are negligible.*

$$p(W|Z_t) \approx p(W|z_{ij}, Z_t)$$

We only use the Assumption 1 when computing the acquisition function. We still perform standard EM during the label aggregation to compute $W$ as in Eq. 2.

With Assumption 1, we express the annotation likelihood by using the estimated worker skills from the last step $p(z_{ij}, Z_t | Y, \hat{W})$. In the DS model, every annotation is independent of each other, and the annotation $z_{ij}$ only depends on its latent ground truth $y_i$ and worker skills $\hat{w}_j$. Therefore, we have $p(z_{ij}, Z_t | Y, \hat{W}) = p(z_{ij} | y_i, \hat{w}_j) p(Z_t | Y, \hat{W})$. We refer to the additional term $p(z_{ij} | y_i, \hat{w}_j)$ as *local* influence since it only affects the $i^{th}$ label.

The added annotation $z_{ij}$ can lead to a larger or more accurate dataset and improve the online-learned ML model accordingly. Since we use the predictions of the ML model as the prior for the ground truths, the improved ML model affects all the labels $Y$. We refer to the change of the ML model as *global* influence.

We ground the local and global influence of adding a single annotation $z_{ij}$ in the label aggregation step in the following:

$$p(Y|z_{ij}, Z_t, X, \hat{W}) \underset{\sim}{\propto} \underbrace{p_\theta(Y|X)}_{\text{global influence}} \underbrace{p(z_{ij}|y_i, \hat{w}_j)}_{\text{local influence}} p(Z_t|Y, \hat{W})$$

$$\theta \leftarrow \arg\min_{\theta'} \mathcal{L}(x_i, p(y_i|z_{ij}, Z_t, x_i), \theta') + \sum_{\bar{i} \neq i} \mathcal{L}(x_{\bar{i}}, p(y_{\bar{i}}|Z_t, x_{\bar{i}}), \theta') \tag{4}$$

From Eq. 4, local influence only affects individual label $y_i$, while global influence affects all labels $Y$. In the next section, we propose LA-BALD, which balances local and global influences brought by adding the annotation $z_{ij}$.

### 4.2 LA-BALD: LABEL AGGREGATION BALD

We propose an information-theoretic acquisition function to explicitly consider the local and global influence in the label aggregator. The prior art Gal et al. (2017) measures the mutual information between ground truth and model parameters $\mathbb{I}(y_i; \theta)$ as the acquisition function in active learning. In image labelling, we obtain noisy annotations from workers. Therefore, we are interested in the mutual information between the annotation and the model parameters $\mathbb{I}(z_{ij}; \theta | X, Z_t, \hat{W})$. We will ignore the conditional variables $X, Z_t, \hat{W}$ in the following for clarity.

We interpret the mutual information as the entropy reduction after conditioning on the other variable, so $\mathbb{I}(z_{ij}; \theta)$ tells us the entropy reduction of the model parameters $\theta$ after conditioning on the annotation $z_{ij}$, corresponding to the global influence in Sec. 4.1.

As described in Sec. 4.1, adding an annotation affects both locally and globally. Therefore, we consider an additional term measuring the mutual information between the annotation and the individual label, $\mathbb{I}(z_{ij}; y_i)$. However, directly summing up $\mathbb{I}(z_{ij}; \theta)$ and $\mathbb{I}(z_{ij}; y_i)$ overestimates the influence gain from the annotation $z_{ij}$ as shown in Fig. 3.a. Instead, we modify the local influence to be $\mathbb{I}(z_{ij}; y_i | \theta)$ which measures conditional entropy reduction of $y_i | \theta$ after conditioning on the annotation $z_{ij}$. The acquisition function of LA-BALD becomes:

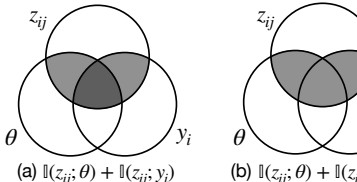

Figure 3: **Venn Diagram between the annotation $z_{ij}$, the individual label $y_i$ and the model parameters $\theta$.** (a) Directly summing up $\mathbb{I}(z_{ij};\theta)$ and $\mathbb{I}(z_{ij};y_i)$ overestimates the information gain from the annotation $z_{ij}$. (b) LA-BALD explicitly considers the local and global influence that the annotation $z_{ij}$ affects to the labels. $\mathbb{I}(z_{ij};\theta)$ is in charge of the global influence, while $\mathbb{I}(z_{ij};y_i|\theta)$ measures the local influence.

| Dataset | #classes | #images | Worker Acc. |
|---|---|---|---|
| Dog | 19 | 18339 | 0.43 |
| Insect+Fungus | 13 | 13000 | 0.65 |
| Vertebrate | 18 | 18000 | 0.72 |
| Commodity | 16 | 16000 | 0.76 |
| Place | 15 | 15000 | 0.87 |
| Dog+Vertebrate | 37 | 49339 | 0.59 |

Figure 4: **Statistics of ImageNet100 sub-tasks.** ImageNet100-sandbox provides sub-tasks with various difficulties and a realistic worker simulation.

$$S_{\text{LA-BALD}}(i,j) = \mathbb{I}(z_{ij};\theta) + \mathbb{I}(z_{ij};y_i|\theta) \tag{5}$$

Later in this section, we discuss the biases introduced by $\mathbb{I}(z_{ij};\theta)$ and $\mathbb{I}(z_{ij};y_i|\theta)$.

### 4.3 BIAS INTRODUCED BY $\mathbb{I}(z_{ij};\theta)$

Dissecting the bias introduced in $\mathbb{I}(z_{ij};\theta)$ and $\mathbb{I}(z_{ij};y_i|\theta)$ helps us understand the interactions between local and global influence. To investigate which image-worker pairs are preferred by $\mathbb{I}(z_{ij};\theta)$, we compare it with BALD scores in Eq. 3. $\mathbb{I}(y_i;\theta)$ measures the entropy reduction of the parameters $\theta$ after conditioning on the ground truth $y_i$.

**Proposition 1.** $\mathbb{I}(y_i;\theta)$ *is the upper bound of* $\mathbb{I}(z_{ij};\theta)$, *if* $z_{ij} = w_j^\top y_i$, *where* $w_j$ *is the confusion matrix representing the* $j^{th}$ *worker skills.*

*Proof.* With slight overloaded notations, $y_i$ is an one-hot vector over $K$ classes and $z_{ij}$ is a probability distribution over $K$ class. With $z_{ij} = w_j^\top y_i$, from the data process inequality Beaudry & Renner (2011), we have the following inequality:

$$\mathbb{I}(z_{ij};\theta) = \mathbb{I}(w_j^\top y_i;\theta) \leq \mathbb{I}(y_i;\theta)$$

The *equality* holds if $w_j$ is an invertible function. $\square$

Proposition 1 tells us that $\mathbb{I}(z_{ij};\theta)$ is maximized when $\mathbb{I}(y_i;\theta)$ is maximized and $w_j$ is invertible. $\mathbb{I}(y_i;\theta)$ tells us how much information gain we can have when obtaining the ground truth of the $i^{th}$ image, and an invertible worker $w_j$ provides noiseless annotations. These two elements are aligned with the intuition to have more valuable and reliable annotations in image labelling.

### 4.4 BIAS INTRODUCED BY $\mathbb{I}(z_{ij};y_i|\theta)$

To investigate which image-worker pairs are preferred by $\mathbb{I}(z_{ij};y_i|\theta)$, we first bring back the ignored conditional variables $Z_t, \hat{W}$ and we have:

**Proposition 2.** *If the label prior* $p(y_i)$ *is a uniform distribution,* $\mathbb{I}(z_{ij};y_i|\theta, Z_t, \hat{W})$ *can be expressed as follow:*

$$\mathbb{I}(z_{ij};y_i|\theta, Z_t, \hat{W})$$
$$= \mathbb{E}_{z_{ij}}[\sum_{y_i} p(y_i|z_{ij},\theta, Z_t, \hat{W}) \log p(z_{ij}|y_i,\hat{w}_j) - \frac{1}{K} \log p(z_{ij}|y_i,\hat{w}_j)]$$

*Proof.* We provide the full derivation in the appendix. $\square$

From proposition 2, we express $\mathbb{I}(z_{ij}; y_i | \theta, Z_t, \hat{W})$ into two terms **i)** minus cross entropy between the annotation likelihood $p(z_{ij}|y_i, \hat{w}_j)$ and the label posteiror $p(y_i|z_{ij}, \theta, Z_t, \hat{W})$ **ii)** cross entropy between the annotation likelihood $p(z_{ij}|y_i, \hat{w}_j)$ and the uniform prior. The image-worker pairs $(i, j)$ provides higher local influences when 1) the annotation $z_{ij}$ is different from the current posterior belief and 2) the $j^{th}$ worker has a higher probability of providing the annotation with higher certainty.

## 5 EXPERIMENTS

In this section, we show the efficacy of LA-BALD compared to the baselines and analyze the behavior of LA-BALD. We first introduce the image annotation benchmark, ImageNet100-sandbox, in Sec. 5.1 and detail the baselines in Sec. 5.2. Then, in Sec. 5.3, we show that LA-BALD outperforms the baselines on most datasets, showing the importance of the additional term $\mathbb{I}(z_{ij}; y_i | \theta)$. In Sec. 5.4, we dissect if the improvements of LA-BALD come from having more accurate annotations or a better model. We provide additional analysis visualizing the interactions between the local and global influence through the annotation time in the appendix.

### 5.1 IMAGENET100-SANDBOX

Evaluating and ablating multi-class image annotation efficiency at scale requires large-scale datasets with diverse and relatively clean labels. We adopt the image annotation benchmark, ImageNet100-sandbox [2] Liao et al. (2021). Some image classification datasets, e.g., Snapshot Serengeti Swanson et al. (2015), provide raw annotations of each worker, potentially allowing us to simulate the labelling process. However, most of the workers in Snapshot Serengeti only provide less than 10 annotations, making it hard to fit into our experimental setting.

ImageNet100-sandbox uses 100 classes sampled from the ImageNet label set Tian et al. (2020) and constructs smaller sub-tasks of varying difficulty. In addition, ImageNet100-sandbox provides realistic worker simulations initialized by the offline crowdsourced worker information. We simulate 50 crowd workers and 10 domain experts. Each worker is represented as a confusion matrix over the label space. The annotation $z_{ij}$ is drawn from the $y_i^{th}$ row of the $j^{th}$ worker's confusion matrix. We summarize ImageNet100-sandbox in Table 4.2 and provide more simulation detail in the appendix.

### 5.2 BASELINES

We categorize our baselines into sequential samplers (SS) and joint samplers (JS). SS compute the scores of images and workers separately, i.e., $S_{\text{image}}(i)$ and $S_{\text{worker}}(j)$, while JS treat them as a whole $S(i, j)$. Depending on which dimension gets sampled first, we categorize SS into image-centric SS and worker-centric SS.

For SS, we adopt image-centric SS (image $\rightarrow$ worker) since worker-centric SS (worker $\rightarrow$ image) might overlook specific hard images. We use the following as our JS baselines: **Rand-I+Rand-W.** Randomly select the image and worker. **Rand-I+Greedy-W.** Randomly select an image and choose greedily the worker based on the estimated worker skills. **Ent-I+Rand-W.** Select the image with the highest entropy in label distribution and randomly select the worker.'' **Ent-I+Greedy-W.** Select the image with the highest entropy in label distribution and greedily select the worker based on the estimated worker skills.

For JS, we adopt the BALD Gal et al. (2017); Houlsby et al. (2011) in Eq. 3, originally proposed in active learning, to image labelling. The acquisition function of **J-BALD** is $S_{\text{J-BALD}}(i, j) = \mathbb{I}(z_{ij}; \theta)$, which only considers the global influence. A similar idea resonates around a decade ago in Yan et al. (2012a), but they do not consider the label aggregator with both local and global influence.

### 5.3 LABELING EFFICIENCY IN IMAGENET100-SANDBOX

For all experiments, we refer to "label accuracy" as the final labels accuracy, "machine accuracy" as the accuracy of the online learning model in Eq. 2, and "annotation accuracy" as $z_{ij}$ accuracy. We

---

[2]https://github.com/fidler-lab/efficient-annotation-cookbook

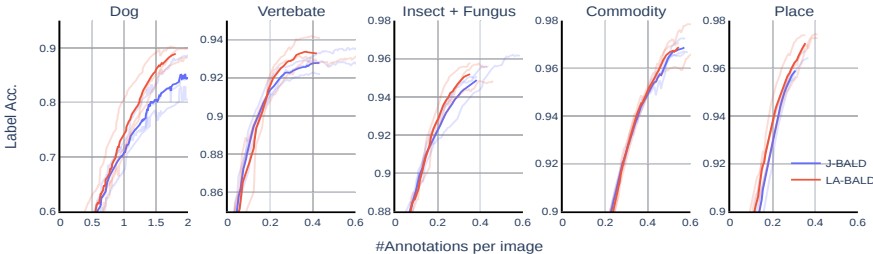

Figure 5: **Image Labeling Efficiency on ImageNet100-sandbox.** We compare LA-BALD with J-BALD, inspired by Gal et al. (2017); Houlsby et al. (2011). LA-BALD reduces the number of annotatiosn by 5% to 37% compared to J-BALD.

run 3 different random seeds in all experiments. We use high-opacity lines to represent the average of different runs and low-opacity lines to represent each run.

We first compare LA-BALD with the JS baseline in Fig. 5 and compare LA-BALD with the SS baselines in Fig. 6. We run over three random seeds for every run and report the average. We show each run with lower opacity at the back. If the different runs end at different annotations, we only report the average when all runs have sample points.

**LA-BALD vs. J-BALD.** As demonstrated in Sec. 4.2, J-BALD considers the change of the ML model but ignores the change of the individual annotation likelihood. In Fig. 5, we show that LA-BALD outperforms J-BALD in *Dog*. To be more specific, LA-BALD reduces the number of annotations by 26% compared to J-BALD when reaching 85.2% label accuracy. LA-BALD also gives consistent improvements in other sub-tasks, reducing the number of annotations by 5% to 37% compared to J-BALD, *Vertebrate* (-37%), *Insect+Fungus* (-21%), *Commodity* (-5%), and *Place* (-6%).

To further understand when we can expect more significant improvements, we compute the average of the top 10 pairs of worker confusion in each task and find that LA-BALD provides more significant improvements in the sub-tasks with higher average confusions, e.g., *Dog* (0.31), *Vertebrate* (0.28) and *Insect+Fungus* (0.24). On the other hand, the improvements are smaller in the sub-tasks with lower average confusions, e.g., *Commodity* (0.16) and *Place* (0.12).

Another reason for the superiority of LA-BALD is that LA-BALD tends to sample the images with higher worker confusion later in the labelling process. Image classes with higher worker confusions, e.g., fiddler crab and rock crab, tend to confuse the online-learned ML model since the ML model is trained on the worker annotations. When the number of annotations is small, the noises from the worker confusion lead to overconfident model predictions and therefore decrease the labelling efficiency. To validate if J-BALD tends to sample the images with high confusion in the early stage of the labelling process, we compute the probability that the images from the most confusing class in each sub-task are sampled in the first half of the labelling process. We find that J-BALD is on average 6% more probable to sample images from the most confusing class than LA-BALD in *Dog* (+2%), *Vertebrate* (+11%) and *Insect+Fungus* (+6%).

**LA-BALD vs. SS.** We show that LA-BALD brings consistent improvements compared to the SS baselines across different sub-tasks. Fig. 6 shows that LA-BALD reduces the number of annotations by 12% on average compared to the second-best SS baselines in each sub-task, ranging from 3% to 20%, *Dog* (-5%), *Vertebrate* (-18%), *Insect+Fungus* (-20%), *Commodity* (-2%), and *Place* (-13%). Note that prior work Liao et al. (2021) adopts **Rand-I+Rand-W** (black curves in Fig. 6). LA-BALD almost halves the number of annotations required in Liao et al. (2021).

**Larger Datasets.** To see if LA-BALD can scale to larger datasets, we conduct experiments on a larger dataset containing 37 classes, *Dog+Vertebrate*. Fig. 7 shows that LA-BALD outperforms all the baselines in terms of labelling efficiency, reducing the number of annotations by 30% compared to J-BALD and 14% compared to Ent-I+Greedy-W.

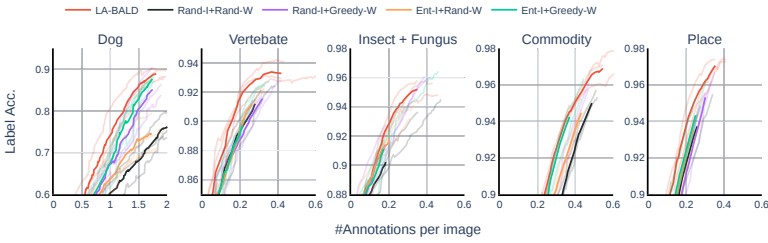
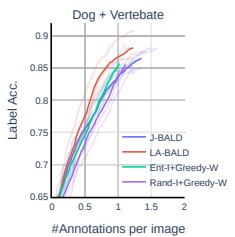

Figure 6: **Comparison of LA-BALD and Sequential Samplers.** LA-BALD outperforms the sequential samplers (SS) in every sub-task, reducing 12% number of annotations compared to the second-best SS baselines in each sub-task.

Figure 7: **Comparison on larger sub-task.** We compare the labeling efficiency of LA-BALD and the baselines.

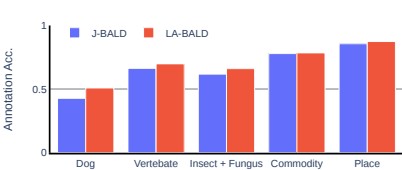

Figure 8: **Annotations Accuracy of J-BALD and LA-BALD.** With similar costs, the accuracy of the annotations sampled by LA-BALD is 6% on average better than that sampled by J-BALD.

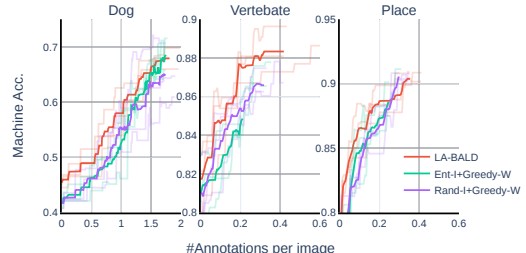

Figure 9: **Comparison of the ML Model Accuracy.** The model trained on the annotations collected from LA-BALD learns quicker than the models trained on the annotations sampled from SS.

## 5.4 MORE ACCURATE ANNOTATIONS OR BETTER MODEL?

We analyze if the improvement of LA-BALD comes from having more accurate annotations or a better model.

**LA-BALD vs. J-BALD.** Fig. 8 shows that the annotations sampled by LA-BALD are 6% better than the annotations sampled by J-BALD on average, showing that LA-BALD is better at choosing relevant image-worker pairs compared to J-BALD. On the other hand, both of them achieve similar ML model performances. To summarize, LA-BALD obtains similar performant ML models compared to J-BALD but is better at choosing relevant image-worker pairs, leading to better label accuracy.

**LA-BALD vs. SS.** Since most sequential samplers tend to stop at early steps and, therefore, only annotate a small set of images, comparing the average annotation accuracy does not provide a fair comparison. Instead, we compare the ML model performance along the time axis in Fig. 9. We ignore the SS baselines with random worker selection for clarity. The model learns quickly in the early stage when trained on the annotations collected from LA-BALD. Please refer to the appendix for a complete comparison of the model accuracy. To summarize, LA-BALD quickly obtains better ML models, leading to better label accuracy.

## 6 CONCLUSION

This work presents LA-BALD, which explicitly considers the local and global influences identified in the state-of-the-art label aggregator. Our studies find that LA-BALD reduces the number of annotations by 19% compared to the common sampler baseline, J-BALD, and 12% compared to the sequential sampler baselines. Our analysis shows that LA-BALD provides more accurate annotations and a better online-learned predictive model, leading to better labelling efficiency over the baselines.

## 7 REPRODUCIBILITY STATEMENT

To implement LA-BALD, we heavily rely on MC dropout (explained in Sec. refbackground:bald and the appendix). For the additional term $\mathbb{I}(z_{ij}; y_i|\theta)$, we follows the BatchBALD Kirsch et al. (2019) implementation. We first separate the second term into $\mathbb{I}(z_{ij}; y_i|\theta) = \mathbb{H}(z_{ij}|\theta) - \mathbb{H}(z_{ij}|y_i, \theta)$. The challenge is to sample $p(z_{ij}|y_i, \theta) = p(z_{ij}|y_i)p(y_i|theta)$. Following the prior work Kirsch et al. (2019), we sample the second term via MC dropout. We release the code in the anonymized repository at https://anonymous.4open.science/r/LA-BALD-8B7D/README.md

## 8 ETHICS STATEMENT

Our work could be applied to many computer vision applications, especially image classification problems requiring high expertise. For example, it could reduce the number of annotations required to train a fine-grained bird classification model used in mobile apps. Our method actively allocates different images to workers, favouring workers with higher expertise. Many reasons lead to the lack of expertise, such as lack of background knowledge, motivations, and familiarity with the tool. However, expertise change over time. Since our proposed method favours workers with higher expertise, it could fail to assign the tasks to new incoming workers, who show low expertise due to the lack of familiarity with the tool. This leads to an imbalanced workload between workers. Expert workers overwork, while workers with low expertise lose their jobs.

Encouraging a balanced workload is essential in practice since it implies fair job recruitment and better parallelism. We can adopt a $\epsilon$-greedy sampling strategy to encourage a balanced workload with LA-BALD. For each step, the task sampler performs random sampling with probability $\epsilon$; otherwise, it performs greedy sampling.

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
