# OpenReview forum: "LA-BALD: An Information-Theoretic Image Labeling Task Sampler"
_ICLR.cc/2023/Conference — Submitted to ICLR 2023_

### Official Review · Reviewer_B5qt · 2022-10-24

**Confidence:** 4
**Correctness:** 4
**Technical Novelty And Significance:** 2
**Empirical Novelty And Significance:** Not applicable
**Recommendation:** 6

**Clarity, Quality, Novelty And Reproducibility:**

English can be much improved in the text, and a revision should consider proofreading by someone proficient in English. Although there are some typos/grammar errors, this paper is easy to follow. However, the technical novelty of the proposed method is somewhat marginal, due to that the authors employ the Liao et al. (2021)’s label aggregator in a straightforward manner.

**Strength And Weaknesses:**

Strengths:

Extensive experimental results demonstrate the effectiveness of the proposed method.

Weaknesses:

1) The novelty in the proposed contributions (employing the label aggregator) to solve the issue is somewhat limited and incremental. More discussions and analyses are required to highlight the contribution of the proposed method.

2) It seems that the proposed method heavily relies on its assumptions (e.g., Assumption 1). This makes the paper of low interest, especially to those expert readers who may not be necessarily working on image classification but other vision problems.

3) The authors do not provide a detailed complexity analysis and efficiency comparison.

**Summary Of The Paper:**

Briefly, to efficiently reduce the noise in the human annotations and improve the predictive model, this paper introduces an image labeling task sampler by actively selecting image-worker pairs according to the formulated information maximization problem, i.e., improving the quality and benefit of image-work pairs for the predicted model. The experimental results are strong.

**Summary Of The Review:**

This paper requires significant improvements. Please address my concerns. I will give a higher rating if most of them are well-addressed.

---

### Official Review · Reviewer_f1dh · 2022-10-26

**Confidence:** 5
**Clarity, Quality, Novelty And Reproducibility:** 1. The presentation could be better. …
**Correctness:** 3
**Technical Novelty And Significance:** 1
**Empirical Novelty And Significance:** 1
**Recommendation:** 3

**Details Of Ethics Concerns:**

N.A.

**Strength And Weaknesses:**

S1. LA-BALD reduces the number of annotations by 19% and 1% on average compared to two baselines.

W1. LA-BALD = LA + BALD. Neither LA (Liao 2021) nor BALD (Gal 2017) was novel.

W2. Motivations of using mutual information or BALD were NOT clear.

W3. Experimental design was relatively simple and did NOT bring more information except direction comparisons.

**Summary Of The Paper:**

This paper proposed LA-BALD for crowdsource labeling of classification datasets. Experiments demonstrated the superiority of LA-BALD over two previous baselines by reducing 19% and 12% of annotation cost.

**Summary Of The Review:**

This paper targeted at a practically important problem of efficient active crowdsource annotation, and proposed an effective approach based on local aggregation framework and Bayesian active learning by disagreement.

However, I expect 1) more novel ideas 2) better experimental designs 3) clearer writing for recommendation of acceptance.

---

### Official Review · Reviewer_Dx5T · 2022-11-01

**Confidence:** 4
**Correctness:** 3
**Technical Novelty And Significance:** 3
**Empirical Novelty And Significance:** 2
**Recommendation:** 5

**Clarity, Quality, Novelty And Reproducibility:**

- Novelty:
    - The proposed LA-BALD not only learns from the worker annotations but also combines with worker annotations in a probabilistic manner.
    - In image labeling, the target goal is to increase the label accuracy, not the accuracy of a novel test set.
    - In contrast to prior work(Yan et al. (2012b)), the LA-BALD considers local and global influences.
- Clearity
    - The connection between Proposition 1 and 2 is unclear to me. In particular, the author stated the point of Proposition 2. However, the explanation is not enough to clearly understand it ("an image-worker pair is favored when the annotation likelihood disagrees with the label posterior, and the uncertainty of \hat{w}_j is small").
    - To support the points, including Appendix of Fig 10., the author could show the empirical evidence with z_ij = w_j^T y_i and the disagreement that an image-worker pair is favored.
    - The author showed robustness to imbalanced dataset learning in the appendix. To support the strength of the LA-BALD, class-wise comparison with baseline (J-BALD) will be helpful to understand better.

**Details Of Ethics Concerns:**

None.

**Strength And Weaknesses:**

- (+) The LA-BALD maximizes the expected information contributing to the labeled dataset via local and global influence by choosing relevant image-worker pairs.
    - The local influence is associated with the individual label.
    - The global influence relates to the online-learned predictive model.
    - The author showed several relevant application results - Bayesian active learning, Imbalanced Learning, and Neural Network Learning. However, there seem to be not enough comparison results.
- (-) As the authors stated, computing the scores of every image-worker pair is time-consuming.
- (-) There are no evaluations on semi-supervised works.

**Summary Of The Paper:**

- The authors proposed an image labeling task sampler that actively selects image-worker pairs to reduce the noise in human label annotations and improve the predictive model.
- The image labeling task sampler is based on an information-theoretic (Label Aggregation BALD, LA_BALD) to maximize the dependencies of the label datasets and the model parameters.
- The experiments on ImageNet100-sandbox showed that the proposed LA-BALD reduces the number of annotations by 19% and 12% on average compared to the two baselines.

**Summary Of The Review:**

- I summarized above.
- There are unclear points to follow Propositions 1 and 2. It would be clear if empirical observations or ablation studies on those points were prepared.

---

### Official Review · Reviewer_vATu · 2022-11-04

**Confidence:** 4
**Clarity, Quality, Novelty And Reproducibility:**
**Correctness:** 3
**Technical Novelty And Significance:** 3
**Empirical Novelty And Significance:** 3
**Recommendation:** 6

**Strength And Weaknesses:**

The major contribution of this work is to extend  bayesian active learning by disagreement to image labeling and incorporates the label aggregator. The combination on improving labeling efficiency and solving local and global influences of individual annotation is interesting. However, there are still some questions that need to be explained.
Questions and detailed comments:
1. What is the motivation to consider BALD to imgae labeling and the difference with the framework motioned by this paper [A].
2. As mentioned in figure 3, the summing up $\mathbb{I}\left(z_{i j} ; \theta\right)$ and $\mathbb{I}\left(z_{i j} ; y_{i}\right)$overestimates the information gain from the annotation $z_{i j}$. If the local influence is not considered, what changes will this overestimates？Please describe in more detail the importance of local influence.
3. The details of the dataset is missing like the statistics and descriptions of the noisy label it contains. In addition to ImageNet100-sanbox, are there additional datasets for model evaluation? It will be hard for readers to catch up how challenging the dataset is and how effective the model is.
4. The new-built baselines are all based on BALD and lack other public baselines on the ImageNet100-sanbox dataset. is there a lack of comparative experiments for method [A]? This comparison result does not clearly show and needs to be described in detail.
[A] Yuan-Hong Liao, Amlan Kar, and Sanja Fidler. Towards good practices for efficiently annotating large-scale image classification datasets. In Proceedings of the IEEE/CVF Conference on Computer Vision and Pattern Recognition (CVPR), pp. 4350–4359, June 2021.

**Summary Of The Paper:**

This paper introduces information-theoretic task sampler, Label Aggregation BALD (LA-BALD) that actively selects image-worker pairs to maximize the information contributing to the labeled dataset via human annotations and the model, and efficiently reduce the noise in the human annotations and improve the predictive model.

**Summary Of The Review:**

Although there are some questions, the paper proposes an interesting and effective sampler for improving labeling efficiency, and also provides some theoretical analysis and detailed formula derivation.

---

### Decision · Program_Chairs · 2023-01-20

**Decision:**

Reject

**Justification For Why Not Higher Score:**

After the reviews and rebuttal this article does not pass the threshold to be published as it is.

**Justification For Why Not Lower Score:**

N/A

**Metareview: Summary, Strengths And Weaknesses:**

This paper introduces an information-theoretic task sampler, Label Aggregation BALD (LA-BALD), that actively selects image-worker pairs to maximize the information contributing to the labeled dataset via human annotations and the model, and efficiently reduce the noise in the human annotations and improve the predictive model. Using BALD for active learning is the way to go since it is state-of-the-art.

However, using BALD for active learning is not novel. The article misses references to previous methods doing a similar thing and comparing with them. There is missing information on the datasets, which makes it difficult to compare with other methods. Computing the scores is time-consuming—no comparisons to semi-supervised. Clarification is needed on equations, more experiments, and ablation studies. LA-BALD = LA + BALD. Neither LA (Liao 2021) nor BALD (Gal 2017) was novel. Motivations of using mutual information or BALD were NOT clear. The experimental design was relatively simple and did NOT bring more information except direction comparisons.

The authors partially answered the requests. However, this article does not pass the threshold to be published as it is.